# Synovial Tissue Proteins and Patient-Specific Variables as Predictive Factors for Temporomandibular Joint Surgery

**DOI:** 10.3390/diagnostics11010046

**Published:** 2020-12-30

**Authors:** Mattias Ulmner, Rachael Sugars, Aron Naimi-Akbar, Nikolce Tudzarovski, Carina Kruger-Weiner, Bodil Lund

**Affiliations:** 1Unit of Cranio- and Maxillofacial Surgery, Karolinska University Hospital, 171 76 Stockholm, Sweden; 2Department of Dental Medicine, Karolinska Institutet, 141 04 Huddinge, Sweden; rachael.sugars@ki.se (R.S.); aron.naimi-akbar@mau.se (A.N.-A.); nikolce.tudzarovski@ki.se (N.T.); carina.kruger-weiner@sll.se (C.K.-W.); Bodil.Lund@uib.no (B.L.); 3Health Technology Assessment-Odontology, Malmö University, 205 06 Malmö, Sweden; 4Department of Oral and Maxillofacial Surgery, Folktandvården Stockholm, Eastmaninstitutet, 113 24 Stockholm, Sweden; 5Department of Clinical Dentistry, Faculty of Medicine, University of Bergen, 5020 Bergen, Norway; 6Department of Oral and Maxillofacial Surgery, Haukeland University Hospital, 5021 Bergen, Norway

**Keywords:** temporomandibular joint, surgery, synovial tissue, synovitis, interleukin, lumican, matrix metalloproteinases, tissue inhibitor of metalloproteinases, cytokine, biomarker

## Abstract

Our knowledge of synovial tissues in patients that are scheduled for surgery as a result of temporomandibular joint (TMJ) disorders is limited. Characterising the protein profile, as well as mapping clinical preoperative variables, might increase our understanding of pathogenesis and forecast surgical outcome. A cohort of 100 patients with either disc displacement, osteoarthritis, or chronic inflammatory arthritis (CIA) was prospectively investigated for a set of preoperative clinical variables. During surgery, a synovial tissue biopsy was sampled and analysed via multi-analytic profiling. The surgical outcome was classified according to a predefined set of outcome criteria six months postoperatively. Higher concentrations of interleukin 8 (*p* = 0.049), matrix metalloproteinase 7 (*p* = 0.038), lumican (*p* = 0.037), and tissue inhibitor of metalloproteinase 2 (*p* = 0.015) were significantly related to an inferior surgical outcome. Several other proteins, which were not described earlier in the TMJ synovia, were detected but not related to surgical outcome. Bilateral masticatory muscle palpation pain had strong association to a poor outcome that was related to the diagnoses disc displacement and osteoarthritis. CIA and the patient-reported variable TMJ disability might be related to an unfavourable outcome according to the multivariate model. These findings of surgical predictors show potential in aiding clinical decision-making and they might enhance the understanding of aetiopathogenesis in TMJ disorders.

## 1. Introduction

Temporomandibular joint (TMJ) diseases might be painful and restrictive by nature, hampering dietary intake and with a negative impact on psychosocial well-being [1]. Surgery is often not considered before a substantial period of failing non-invasive treatments has been tried. From this perspective, the demands on surgery from the affected patients are higher, which accounts for the long duration of symptoms and it is regarded as chronic at this timepoint. Arthroscopy is a minimally invasive surgical alternative that is often used in cases of disc displacement (DD), osteoarthritis (OA), and chronic inflammatory arthritis (CIA) [2,3,4]. Discectomy is an open surgical procedure that is mainly used for DD [5,6]. The outcome of arthroscopy or discectomy, when applied to patients with DD, OA, and CIA, has been reported to be 60 to 88%, where open joint surgery seems to be slightly superior when compared to arthroscopy in a meta-analysis [2,3,4,5,6,7]. Inferior surgical outcome can be prevented by examining the patient in an organised fashion and applying strict diagnostic criteria in search of the correct diagnosis [8,9]. This is the foundation for the surgical decision, but, since TMJ DD, as well as TMJ OA, still lack formal explanatory grounds, a better understanding of the aetio-pathogenesis might shed new light on both diagnostic criteria and best-practice treatment. Characterising the synovial tissue profile and identifying patient-specific predictive factors is a possible approach for enhancing the surgical outcome. This will benefit the patient, as well as regulatory authorities, since a good surgical outcome often prevents further treatment, reduces medication, and minimises sick leave.

Potential predictive factors for TMJ surgery, such as age, TMJ pain, maximal interincisal opening (MIO), psychiatric co-morbidity, and masticatory muscle palpation pain, have been investigated [2,10,11,12,13]. In addition, cytokines have been identified in the TMJ synovial fluid, and high concentrations of interleukin (IL) 10 have been proposed for predicting a successful surgical outcome [14]. Studies have already highlighted cytokine localisation in synovial tissue as a valuable biomarker and predictor for treatment in rheumatoid arthritis [15,16]. However, this has not been assessed for the TMJ and diagnoses that are associated with TMJ disease or disorder. 

The primary aim of the present study was to investigate synovial tissue protein concentrations and relate them to surgical outcome. The hypothesis was that the concentrations of pro-inflammatory cytokines were higher in patients with inferior outcome, whilst the anti-inflammatory cytokines were higher in patients with superior surgical outcomes. The secondary aim was to control for recorded objective and subjective patient variables, and their relation to surgical outcome. Identifying clinical variables or synovial tissue proteins that might influence surgical outcome could be a valuable contribution to oral- and maxillofacial surgeons decision-making. To our knowledge, this is the first attempt to investigate TMJ synovial tissue proteins in relation to surgical outcome.

## 2. Materials and Methods 

### 2.1. Study Design

A prospective cohort study was performed at the Unit of Cranio- and Maxillofacial Surgery, Karolinska University Hospital, Stockholm, Sweden. The Regional Ethics Review Board approved the study (registration number 2014/622-31/1, approved on 23 April 2014. The patients referred due to DD with reduction (DDwR), DD without reduction (DDwoR), OA together with arthralgia, and CIA were eligible for inclusion. The patients were enrolled from December 2014 to January 2017 and written informed consent was mandatory before inclusion. The study was designed, and the article written, in accordance with the STROBE statement.

### 2.2. Study Population

TMJ diagnoses were set according to the Diagnostic Criteria for Temporomandibular Disorders (DC/TMD), except for CIA diagnosis, where the requirement was rheumatic diagnosis that was set by a rheumatologist [8]. Criterions for surgery and inclusion were that the patient had one of the diagnoses DDwR, DDwoR, OA, or CIA, had tried non-invasive therapy for at least 3–6 months, visual analogue scale (VAS) value of ≥4 for TMJ functional pain or TMJ disability, and that DDwoR patients had a MIO of ≤35 mm. The patients were excluded if they had prior open TMJ surgery, were unable to give informed consent, or were younger than 18 years. 

### 2.3. Clinical Examination

Patient-specific data were registered preoperatively, one and six months postoperatively, while using a standardised case record form. Patient inclusion and data gathering were performed by M.U., A.N.-A., C.K.-W, and B.L., who were calibrated for patient classification and clinical examination. The anamnestic variables collected included present illnesses, medication, prior jaw trauma, ongoing tinnitus/ear fullness affected side, duration of present TMJ symptoms, and subjective grading on a 0–10 graded VAS of TMJ pain, TMJ disability, psychosocial impact of TMJ problems, and global pain [17]. Joint mobility was measured with the Beighton scoring system, and a value of ≥4 was regarded to be indicative of general joint hypermobility [18]. Positive findings of palpation pain of the masticatory muscles and the TMJ’s, and measurements of MIO, lateral excursion, and protrusion were registered in accordance with DC/TMD [8]. A calibration exercise preceded Wilks classification and two of the researchers (M.U. and B.L.) subsequently individually performed the grading [19]. Divergent conclusions were discussed, and consensus was reached. 

Surgical outcome was based on four parameters: MIO, TMJ pain, TMJ disability, and TMJ psychosocial impact registered at the last planned visit six months after surgery, and classified as either successful, good, intermediate, or deteriorated. The criteria for successful treatment were objective measurement of MIO ≥ 35 mm, and all subjective VAS scoring of TMJ pain, TMJ disability, and TMJ psychosocial impact of ≤3 or ≥40% improvement. A good surgical outcome was defined as MIO ≥ 35 mm and whether one or two of the VAS values of pain, functional disabilities, and psycho-social impact showed ≥40% improvement or a VAS value of ≤3. If the above-mentioned criteria got obviously worse, then the outcome was deemed to be deteriorated. With minor or no improvements, the result was classified as intermediate. 

### 2.4. Surgical Procedure and Collection of Tissue Samples

According to the departments’ research-based guidelines, patients with DDwR were scheduled for discectomy, and patients with DDwoR, OA, or CIA had arthroscopic lysis and lavage generally. One surgeon performed all of the operations (M.U.). Two synovial tissue biopsies were taken from the posterior bilaminar zone in the superior joint compartment. The triangulation technique was used in order to collect biopsies under direct visualisation during arthroscopy (Figure 1) [20]. Biopsy forceps (Karl Storz SE & Co, Tuttlingen, Germany) were used, resulting in approximately 4 mm^2^ tissue samples. Synovial tissue samples that were destined for protein extraction were placed in RNAlater (ThermoFisher Scientific, Waltham, MA, USA) and then refrigerated for 24 h. RNAlater was then removed and the samples stored at −80 °C. 

### 2.5. Analysis of Synovial Tissues

Synovial tissue was ground in liquid nitrogen in order to disrupt the tissue piece. The proteins were extracted in ice-cold cell lysis buffer NP40, prepared according to the manufacturer’s instructions (ThermoFisher Scientific) [21]. 50 µL cell lysis buffer per 10 mg of tissue was used. The mixtures were centrifuged at 20,000× *g* at 4 °C for 10 min., and the supernatant stored at −80 °C until analysis.

The total protein concentration in each tissue sample was determined while using the Qubit Protein Assay Kit (ThermoFisher Scientific) and the Qubit Fluorometer (ThermoFisher Scientific). Magnetic bead panels HTMP2MAG-54K, HMMP2MAG-55K, and HCYTOMAG-60K (Merck Millipore, Burlington, MA, USA), and LXSAHM-20 (R&D Systems, Bio-Techne Corp., Minneapolis, MN, USA), were used in order to determine the levels of synovial tissue proteins with multi-analytic profiling while using a Luminex 200 system (Luminex, Austin, TX, USA) and xMAP technology. Attained data were analysed by xPONENT 3.1 software (Luminex). HCYTOMAG-60K and LXSAHM-20 were customised and contained the following proteins: a disintegrin and metalloproteinase with a thrombospondin type 1 motif member 13 (ADAMTS13), aggrecan, bone morphogenetic protein (BMP) 2, 4, and 9, collagen 1-α1, collagen 4-α1, epidermal growth factor (EGF), eotaxin, fibroblast activation protein (FAP), fibroblast growth factor 2 (FGF-2), fibronectin, granulocyte-colony stimulating factor (G-CSF), granulocyte-macrophage (GM) CSF, hepatocyte growth factor receptor (HGFR), intercellular adhesion molecule 1 (ICAM-1), IL-1β, IL-1ra, IL-6, IL-7, IL-8, IL-10, IL-17, interferon gamma-induced protein 10 (IP-10), lumican, monocyte chemoattractant protein 1 (MCP-1), macrophage inflammatory protein 1α (MIP-1α), MIP-1β, neural cell adhesion molecule (NCAM), osteoprotegerin (OPG), osteonectin, platelet-derived growth factor (PDGF) AA, PDGF-AB/BB, regulated on activation normal T-cell expressed and secreted (RANTES), syndecan-1, syndecan-4, tenascin C, transforming growth factor α (TGF-α), tumour necrosis factor α (TNF-α), TNF-β, triggering receptor expressed on myeloid cells 1 (TREM-1), and vascular endothelial growth factor (VEGF). In addition, HTMP2MAG-54K and HMMP2MAG-55K contained matrix metalloproteinase 1 (MMP-1), MMP-2, MMP-7, MMP-9, MMP-10, tissue inhibitor of metalloproteinase 1 (TIMP-1), TIMP-2, TIMP-3, and TIMP-4. Fifty-one proteins were analysed in total. 

### 2.6. Statistical Analyses 

Stata version 15 (StataCorp, Collage Station, TX, USA) and IBM SPSS version 25.0 (IBM Corp, Armonk, NY, USA) were used to analyse the data. The descriptive statistics were calculated as mean ± SD for all continuous data and as a number and percentage for bivariate data. Data on patient characteristics were analysed with Student’s T-test for continuous data and Fisher’s exact test for categorical data. For the statistical analyses of synovial tissue proteins, the surgical outcome groups intermediate and deteriorated were merged into one group, since there was only one patient in the deteriorated group. The concentration of specified proteins (pg/mL) was used in the statistical analyses. The surgical outcome was the dependent variable and ordered logistic regression were used for both univariate and multivariate computations. The multivariate regression model was tested with Akaike’s information criterion (AIC) in order to estimate the performance of the model. The best performance was reached by including the specific protein and the potential confounders CIA, TMJ disability, masticatory muscle palpation, and the interaction of CIA and positive finding of masticatory muscle palpation pain. Masticatory muscle palpation was dichotomised, and no finding of palpation pain was merged with unilateral positive sign, since it made the model perform better according to AIC. A *p*-value of ≤0.05 was regarded as significant.

## 3. Results

### 3.1. Patient Demographics and Patient-Specific Clinical Variables

One-hundred patients had followed the protocol at study closure (Figure 2). The 27 patients who were excluded or did not participate had a mean age of 40.4 years (SD 15.3) and 81% were women. No differences were found when comparing participating patients to the non-participating with regards to sex and age. In six patients out of the hundred included, it was not possible to harvest any synovial tissue; therefore, their data were only included in the clinical parameter analyses. Table 1 compiles demographic data and preoperative patient-specific symptoms and signs. The outcome of surgery, as well as measured mean differences before and after surgery are displayed in Table 2. Patients in the diagnosis-group OA were significantly older when compared to the other groups (*p* = 0.022) and they had more co-morbidities, classified as “other diseases” (*p* = 0.041). Both OA (*p* = 0.003) and DDwR patients (*p* < 0.001) had larger MIO as compared to the rest of the cohort. The group with DDwR had significantly lower TMJ pain VAS-value (*p* = 0.008) and fewer patients with both palpation pain of the masticatory muscles (*p* < 0.001) and the TMJ (*p* < 0.001). On the other hand, the CIA-group had significantly more patients with palpation pain on muscles (*p* = 0.006) and TMJ (*p* < 0.001). Patients with DDwoR had significantly lower TMJ psychosocial impact (*p* = 0.003), as well as lower global pain (*p* = 0.050), when compared to the other three diagnoses.

All of the registered diagnoses and patient-specific factors were tabulated and those showing signs of association to outcome were further analysed in a univariate fashion. TMJ palpation pain (coef., 0.89; *p* = 0.044) and masticatory muscle palpation pain (coef., 1.97; *p* < 0.001) were both positively associated to a worse outcome (Figure 3). The four subjective VAS variables all had a linear association with surgical outcome, which was significant for TMJ disability (coef., 0.29; *p* = 0.011), TMJ psychosocial impact (coef., 0.15; *p* = 0.032), and global pain (coef., 0.13; *p* = 0.043), but not for TMJ pain (coef., 0.16; *p* = 0.073) (Figure 4). Tinnitus, sex, age, MIO, psychiatric disorder, TMJ pain, and the TMJ diagnoses showed no significant correlation to outcome.

### 3.2. Synovial Tissue Analysis, Univariate Analysis

When examining the proteins in the multi-analytic profiling system, some of the proteins were identified as being below or above the standard limits, as defined by the manufacturers. Those samples that were below the lowest standard were set at the lowest standard value and those above the highest standard were set at the highest standard value. The processed tissue samples with protein measurements out of the assay’s precision and recovery were treated as the missing values.

ADAMTS13, BMP-9, HGFR, IL-7, MMP-10, NCAM, osteonectin, syndecan-1 and 4, TIMP-4, and TREM-1 were found with detectable concentrations in most patients. These proteins have not been previously described in the human TMJ.

All of the analysed proteins were related to outcome in a univariate ordered logistic regression model. Higher concentrations of both eotaxin (coef., 2.89 × 10^−3^; *p* = 0.038) and syndecan-1 (coef., 1.11 × 10^−4^; *p* = 0.024) significantly changed the outcome in a negative direction. None of the other proteins had any significant correlation to outcome.

### 3.3. Multivariate Analysis of Synovial Tissue and Potential Confounders

The significant results from univariate analyses with respect to patient-specific variables were tested in a multivariate model. The tested variables were TMJ disability (coef., 0.23; *p* = 0.054), TMJ psychosocial impact (coef., 0.06, *p* = 0.424), global pain (coef., 0.07, *p* = 0.352), and masticatory muscle palpation pain (coef., 1.69; *p* = 0.001). Table 3 presents multivariate ordered logistic regression analyses of association between the outcome of TMJ surgery and the specific proteins, including potential confounders and the interaction between CIA and positive bilateral masticatory muscle palpation pain. Higher concentrations of IL-8, lumican, MMP-7, and TIMP-2 were all associated to an inferior outcome in a significant way. ADAMTS13, BMP-4, eotaxin, NCAM-1, and TIMP-1 were close to significant, with *p*-values of ≤ 0.075. Patients with the interaction CIA and bilateral masticatory muscle palpation pain showed a significant association to a positive surgical outcome in the analysis of ADAMTS13, IL-1β, and TNF-β (Table 3). All of the analyses of the interaction variable showed a negative coefficient, indicating that positive bilateral muscle palpation pain does not predict a poor surgical outcome in patients that are suffering from CIA. 

## 4. Discussion

The success rates in TMJ surgery have been reported as variable and they often not better than 80%. Identifying patient-specific predictors might be a valuable tool for surgeons, patients, and health-care providers to improve outcome. 

The investigation of TMJ synovial fluid proteins potentially reflecting surgical outcome has to our knowledge been done twice before, where higher concentrations of IL-10 were significantly associated with a positive outcome of arthrocentesis, and TMJ pain was associated with higher concentrations of IL-6 and IL-8 indicating a negative outcome [14,22]. The TMJ synovial tissue proteins have not been investigated in relation to outcome earlier. In this study, four proteins—IL-8, lumican, MMP-7, and TIMP-2—were found to be associated with an impaired surgical outcome in a concentration dependent matter in multivariate analyses. The chemokine IL-8 exerts effects on cells, such as fibroblasts, neutrophils, and synovial cells during normal function and with an inflammatory state [23]. Higher levels of IL-8 have been associated with a higher severity of disease in rheumatoid arthritis and when comparing DDwR to DDwoR [23,24]. In oral squamous cell carcinoma, IL-8 was reported to up-regulate the production of MMP-7 via the IL-8 receptor β [25]. MMP´s are a group of proteases with the ability to degrade components of extracellular matrix (ECM) [26,27]. MMP-7 has been found to act on several collagens and proteoglycans directing to its role in joint degradation [26,27,28]. The main endogenous inhibitors of MMPs are TIMPs that bind MMPs in a 1:1 ratio [29]. TIMP-2 has been proposed to serve as a continuous ECM protector. Some of the studies have suggested that its mRNA expression does not respond to different stimuli during basal or inflammatory activity in joints, whereas other studies have detected mRNA in response to osteopontin or relaxin levels [30,31,32]. The small, leucine-rich, proteoglycan lumican has been associated with wound healing and found to be increased in degenerated TMJ discs when compared to normal discs [33,34]. All four proteins with a negative correlation to outcome are related to tissue turnover and remodeling, where lumican and TIMP-2 are suggested to promote TMJ healing, whilst IL-8 and MMP-7 possibly have degenerative properties. Therefore, they may potentially be useful as individual markers for a negative outcome and, if they are also demonstrated to relate to each other, they might provide a protein pattern that is indicative of biomarker quality. 

In the univariate statistical analyses, higher concentrations of eotaxin and syndecan-1 showed a correlation to a suboptimal surgical outcome. Eotaxin is a chemokine that has been shown to increase osteoclast activity in bone inflammation, while syndecan-1 might be associated with attempted cartilage repair [35,36]. Fibrocartilage stem-cells (FCSC) with chondrogenic differentiation abilities have been identified in the human TMJ cartilage [37]. The association between the transmembrane proteoglycan syndecan-1 and FCSC reparative traits is unknown, but it deserves attention. 

The age of the patient and preoperative MIO have previously been described as predictive factors for TMJ surgical outcome [2,13,38,39]. In the current study, this could not be confirmed for MIO and age. Univariate analysis revealed that TMJ disability, TMJ psychosocial impact, global pain, masticatory muscle, and TMJ palpation pain were significantly related to outcome. In the multivariate analysis, only masticatory muscle palpation pain remained significant. TMJ disability and CIA were included in the multivariate model, because they, according to AIC, strengthened the model. CIA showed a significant negative correlation to outcome in 10 of the 51 multivariate analyses of specified proteins, and TMJ disability in six of 51. This might imply that the diagnosis CIA and the variable TMJ disability individually can be valid predictors for a negative outcome of TMJ surgery.

The association between masticatory muscle palpation pain and negative outcome has earlier been presented by our group in two different patient cohorts [2,13]. Considering that the variable was significant in all but four multivariate calculations advocates its potential as a predictive factor. In contrast, CIA patients with the presence of bilateral muscle palpation pain did not seem to have any additional negative impact on outcome. The result strongly suggests that bilateral masticatory muscle palpation pain is an important predictive factor, which is why DDwR-, DDwoR-, and OA-patients with these findings should alert the clinician to consider a new round of non-invasive therapy.

A shortcoming of the study was the loss of 27 eligible patients, who did not participate for different reasons. Potential bias might be considered because the only variables possible to analyse for the non-participant group were sex, age, and TMJ diagnosis. A relatively short follow-up period was used, which might also implicate a bias in some of the diagnostic groups. Because only four out of 51 proteins correlated with outcome, there was a risk that these associations are by chance. This suggests that further investigations should be made to verify these findings.

To conclude, preoperative bilateral palpation pain of the masticatory muscles was found to be a predictor for negative surgical outcome, and it might alert the surgeon to consider non-invasive interventions that have not yet been tried before scheduling surgery. However, in patients diagnosed with CIA, bilateral masticatory muscle pain did not indicate a negative surgical outcome when compared to the other included TMJ diagnoses. TMJ disability was the only outcome measure that showed potential as predicting factor. IL-8, lumican, MMP-7, and TIMP-2 were individually shown to have a positive correlation to worse outcome. Altogether, the results demonstrate that the clinical variable bilateral masticatory muscle palpation pain seems to be a more robust predictor for surgical outcome when compared to any of the investigated proteins. Description of protein alterations due to diagnosis, severity, and progress of TMJ disease, but also in relation to different treatment modalities, has to continue. Further mapping will possibly reveal more of the potential multi-factorial pathogenesis. 

## Figures and Tables

**Figure 1 diagnostics-11-00046-f001:**
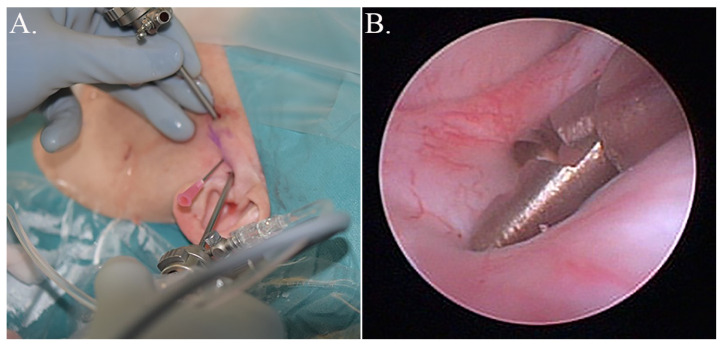
Photographs showing the synovial sample procedure. (**A**) In the triangulation technique, the instrument canal closest to the patient’s ear contained the optic and the second instrument canal the biopsy forceps. (**B**) A synovial tissue sample from the posterior bilaminar zone in the superior temporomandibular joint (TMJ) compartment was taken with the biopsy forceps.

**Figure 2 diagnostics-11-00046-f002:**
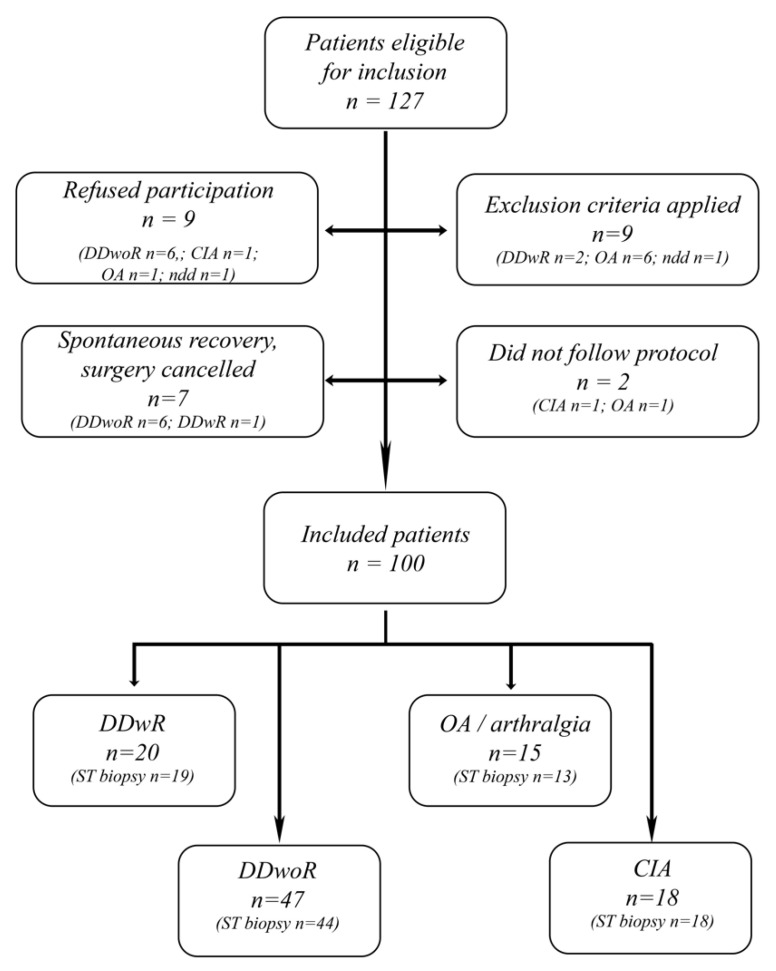
Flow chart illustrating patients´ eligibility for inclusion into the study, reasons for not participating, and TMJ diagnoses. CIA, chronic inflammatory arthritis; DDwoR, disc displacement without reduction; DDwR, disc displacement with reduction; OA, osteoarthritis; *n*, number; ndd, no diagnosis defined; ST, synovial tissue.

**Figure 3 diagnostics-11-00046-f003:**
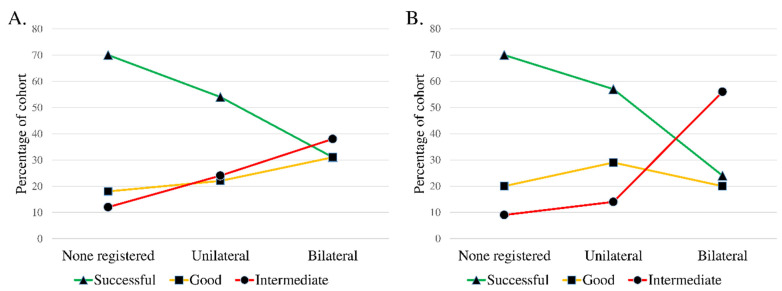
Line charts illustrating the preoperative TMJ and muscle palpation variables related to surgical outcome groups. The intermediate outcome group also contains the single deteriorated patient. Palpation of the lateral aspect of the TMJ and palpation of the masseter and temporal muscle was performed in accordance with the Diagnostic Criteria for Temporomandibular Disorders (DC/TMD). Positive palpation findings were recorded as being unilateral or bilateral. Negative palpation findings were recorded as none registered. (**A**) The line chart shows the significant linear association of increased positive findings of TMJ palpation pain related to a worse outcome (*p* ≤ 0.05). (**B**) Masticatory muscle palpation pain had a strong association to surgical outcome in a similar manner as TMJ palpation pain (*p* ≤ 0.005). DC/TMD, diagnostic criteria for temporomandibular disorders; TMJ, temporomandibular joint.

**Figure 4 diagnostics-11-00046-f004:**
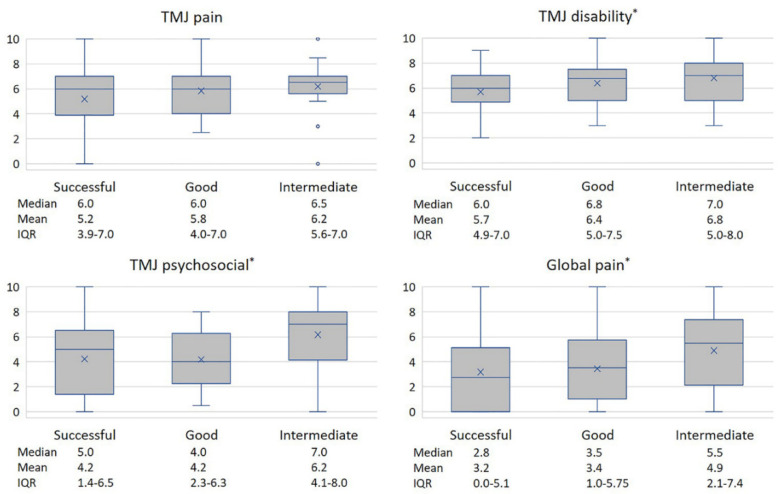
Box plot showing the relation between preoperative patient-reported 0–10 VAS values of TMJ pain, TMJ disability, TMJ psychosocial, and global pain according to surgical outcome groups. The top of the box indicates the 75th percentile, and the bottom the 25th percentile. The line within the box shows the median and the cross indicates the mean. The whiskers show the 10th and 90th percentile and points outside the 10th and 90th percentile shows outliers. All four preoperative VAS values were higher in relation to a worsened outcome, but only TMJ pain were not significant. TMJ, temporomandibular joint; VAS, visual analogue scale. * *p* < 0.05.

**Table 1 diagnostics-11-00046-t001:** Preoperative registration of demographic data, anamnestic information, objective and subjective measurements of included patients.

Classification	DDwR	DDwoR	OA	CIA	Total
Demographic data					
Number of patients	20	47	15	18	100
Sex, W/M	15/5	40/7	15/0	17/1	87/13
Age (years), mean (SD)	37.5 (11.7)	43.1 (15.8)	50.7 (20.3)	38.1 (13.0)	42.2 (15.8)
Patient history					
Duration (mos.) mean (SD)	43.3 (40.4)	20.7 (24.0)	29.3 (44.1)	43.5 (42.7)	30.4 (35.7)
Tinnitus/ear fullness, *n* (%)	5 (25)	14 (30)	5 (33)	5 (28)	29 (29)
TMJ trauma, *n* (%)	8 (40)	10 (21)	3 (20)	3 (17)	24 (24)
Medical history, *n* (%)					
Healthy	10 (50)	19 (40)	2 (13)	0 (0)	31 (31)
Psychiatric disorder	4 (20)	15 (32)	1 (7)	3 (17)	23 (23)
Neuropsychiatric disorder	1 (5)	1 (2)	0 (0)	1 (6)	3 (3)
Autoimmune disease	0 (0)	0 (0)	1 (7)	18 (100)	19 (19)
Metabolic disease	2 (10)	6 (13)	3 (20)	0 (0)	11 (11)
Other disease	6 (30)	22 (47)	11 (73)	10 (56)	49 (49)
Objective measures					
MIO, mm (SD)	43.8 (9.7)	29.2 (4.7)	40.4 (5.7)	31.2 (7.0)	34.1 (8.9)
Wilks classification, mean (SD)	2.6 (0.9)	3.9 (0.6)	na	na	3.6 (1.0)
TMJ palp pain, *n* (bilat/lat/no)	0/6/14	4/28/15	2/10/3	7/10/1	13/54/33
Muscle palp pain, *n* (bilat/lat/no)	2/4/14	7/10/28	6/3/6	9/3/5	24/20/53
Subjective measures (VAS 0-10), mean (SD)					
TMJ pain	4.3 (2.5)	5.6 (2.4)	6.3 (2.2)	6.2 (1.9)	5.6 (2.4)
TMJ disability	6.1 (2.0)	6.3 (1.7)	5.7 (2.2)	5.8 (1.8)	6.1 (1.8)
TMJ psychosocial impact	5.1 (3.3)	3.7 (2.8)	5.7 (3.6)	5.7 (2.4)	4.6 (3.0)
Global pain	3.1 (3.2)	3.0 (3.0)	4.9 (2.5)	4.8 (2.9)	3.6 (3.0)

Bilat, bilateral; CIA, chronic inflammatory arthritis; DDwoR, disc displacement without reduction; DDwR, disc displacement with reduction; lat, lateral; M, men; MIO, maximum interincisal opening; mos., months; *n*, number; na, not applicable; OA, osteoarthritis; palp, palpation; SD, standard deviation; TMJ, temporomandibular joint; VAS, visual analogue scale; W, women.

**Table 2 diagnostics-11-00046-t002:** Outcome of surgery for the total cohort and for the different TMJ diagnoses, comparing mean differences of preoperative and postoperative values using paired samples t-test.

Outcome	Successful	Good	Intermediate	Deteriorated ^a^
Number of patients (%)				
Total	56 (56)	22 (22)	21 (21)	1 (1)
DDwR	14 (70)	3 (15)	3 (15)	0 (0)
DDwoR	24 (51)	14 (30)	9 (19)	0 (0)
OA	9 (60)	4 (27)	2 (13)	0 (0)
CIA	9 (50)	1 (6)	7 (39)	1 (6)
Preoperative measurements sorted according to outcome				
Sex, W/M	48/8	18/4	20/1	
Age (years), mean (SD)	44.5 (16.0)	39.1 (16.2)	39.4 (14.6)	
Patient history				
Duration (mos.) mean (SD)	30.9 (39.5)	30.0 (31.1)	30.5 (31.8)	
Tinnitus/ear fullness, *n* (%)	13 (23)	7 (32)	9 (43)	
TMJ trauma, *n* (%)	15 (27)	7 (32)	2 (10)	
GJH, *n* (%)	12 (21)	7 (32)	5 (23)	
Medical history, *n* (%)				
Healthy	19 (34)	6 (27)	6 (29)	
Psychiatric disorder	12 (21)	5 (23)	6 (29)	
Neuropsychiatric disorder	0 (0)	2 (9)	1 (5)	
Autoimmune disease	1 (2)	1 (5)	1 (5)	
Metabolic disease	7 (13)	3 (14)	1 (5)	
Other disease	26 (46)	13 (59)	10 (48)	
Objective measures, mean (SD)				
MIO (mm)	34.4 (7.6)	33.9 (11.8)	33.5 (9.3)	
Wilks classification	3.5 (1.0)	3.6 (0.9)	4.0 (0.9)	
Mean differences of pre- and postoperative measurements in relation to outcome				
Objective measures (mm), mean (SD)				
MIO	8.4 (7.2) **	7.1 (8.0) **	0.5 (7.2)	
LTR left	0.3 (2.2)	1.9 (2.8) *	-0.7 (3.0)	
LTR right	0.8 (3.1) *	0.3 (2.8)	0.7 (2.1)	
PTR	1.5 (2.9) **	1.7 (2.6) *	0.6 (2.0)	
Subjective measures (VAS 0-10), mean (SD)				
TMJ pain	−4.1 (2.4) **	−1.6 (2.4) **	−0.7 (1.7)	
TMJ disability	−4.2 (1.8) **	−2.5 (2.2) **	−0.5 (1.9)	
TMJ psychosocial	−3.5 (2.6) **	−0.8 (3.2)	−0.1 (2.6)	
Global pain	−1.2 (2.8) **	−0.3 (2.0)	0.5 (3.3)	

CIA, chronic inflammatory arthritis; DDwoR, disc displacement without reduction; DDwR, disc displacement with reduction; GJH, general joint hypermobility; LTR, lateral excursion; M, men; MIO, maximum interincisal opening; mm, millimetre; mos., months; *n*, number; OA, osteoarthritis; PTR, protrusion; SD, standard deviation; TMJ, temporomandibular joint; VAS, visual analogue scale; W, women. ^a^ The patient with deteriorated outcome was transferred to the intermediate group for statistical analyses. * *p* ≤ 0.05, ** *p* ≤ 0.005.

**Table 3 diagnostics-11-00046-t003:** Ordered logistic regression relating the dependent variable surgical outcome (successful, good, intermediate/deteriorated) to analysed proteins, potential confounders and the interaction of CIA and positive jaw muscle palpation tenderness.

	No. obs.	Specified Protein ^a^	CIA ^b^	Masticatorymuscle Palpation ^b^	TMJ Disability ^b^	Interaction CIA/Palp.^c^
Protein		Coef.	Coef.	Coef.	Coef.	Coef.
ADAMTS13	87	7.29 × 10^−7^	3.78 **	2.15 **	0.27 *	−4.19 *
Aggrecan	94	−8.25 × 10^−6^	1.77	1.93 **	0.22	−2.07
BMP-2	94	−9.06 × 10^−5^	1.24	1.87 **	0.23	−1.69
BMP-4	94	8.05 × 10^−4^	1.34	1.95 **	0.22	−1.62
BMP-9	44	1.45 × 10^−3^	1.00	1.73	0.24	−1.54
Collagen-1 α1	94	4.28 × 10^−6^	1.31	1.90 **	0.21	−1.66
Collagen-4 α1	94	−2.38 × 10^−6^	1.15	1.86 **	0.23	−1.64
EGF	94	4.33 × 10^−4^	1.16	1.86 **	0.22	−1.64
Eotaxin	93	2.69 × 10^−3^	1.19	1.92 **	0.22	−1.61
FAP	94	1.05 × 10^−5^	1.37	1.95 **	0.18	−1.72
FGF-2	94	4.03 × 10^−5^	1.22	1.99 **	0.23	−1.77
Fibronectin	94	5.12 × 10^−8^	1.26	1.83 *	0.20	−1.66
G-CSF	94	−2.32 × 10^−4^	1.82	2.01 **	0.22	−2.37
HGFR	94	6.68 × 10^−5^	1.34	1.92 **	0.21	−1.68
ICAM-1	94	1.54 × 10^−8^	1.17	1.86 **	0.22	−1.65
IL-1β	85	−2.57 × 10^−2^	1.98 *	2.01 **	0.20	−3.27 *
IL-1ra	94	2.51 × 10^−3^	1.24	1.90 **	0.21	−1.64
IL-6	46	8.66 × 10^−3^	1.33	0.91	0.04	−0.55
IL-7	94	−6.60 × 10^−4^	1.54	1.91 **	0.22	−2.01
IL-8	93	2.17 × 10^−2^ *	1.11	2.11 **	0.20	−1.33
IL-10	92	5.66 × 10^−3^	1.56	1.88 *	0.25 *	−2.00
IP-10	94	6.99 × 10^−5^	1.13	1.68 *	0.23	−1.40
Lumican	94	9.99 × 10^−8^ *	1.48	2.02 **	0.17	−1.82
MCP-1	94	−2.43 × 10^−4^	1.09	1.91 **	0.23	−1.53
MIP-1α	49	1.70 × 10^−3^	1.71	1.97	−0.09	−2.80
MIP-1β	60	3.61 × 10^−3^	1.73 *	1.63 *	0.16	−1.70
MMP-1	66	9.72 × 10^−4^	17.74	2.56 **	0.26	− ^d^
MMP-2	93	1.92 × 10^−7^	1.55	1.86 **	0.24	−2.04
MMP-7	62	3.06 × 10^−5^ *	− ^e^	3.22 **	0.23	− ^e^
MMP-9	93	1.92 × 10^−6^	1.61 *	1.90 **	0.24	−2.07
MMP-10	89	−5.65 × 10^−5^	1.57	1.80 *	0.26 *	−2.02
NCAM-1	94	1.74 × 10^−5^	1.33	1.90 **	0.22	−1.58
OPG	94	1.73 × 10^−5^	1.31	1.80 *	0.20	−1.61
Osteonectin	94	2.98 × 10^−8^	1.17	1.86 **	0.22	−1.66
PDGF-AA	94	1.28 × 10^−4^	1.16	1.84 *	0.22	−1.61
PDGF-AB/BB	94	−4.84 × 10^−5^	1.27	1.88 **	0.21	−1.84
RANTES	94	−4.13 × 10^−5^	1.22	1.80 *	0.22	−1.92
Syndecan-1	94	7.62 × 10^−5^	0.60	1.72 *	0.19	−0.98
Syndecan-4	90	9.74 × 10^−5^	1.20	1.89 **	0.19	−1.66
Tenascin C	94	4.13 × 10^−6^	1.23	1.89 **	0.20	−1.70
TIMP-1	93	3.91 × 10^−5^	2.01 *	1.89 **	0.20	−2.04
TIMP-2	93	3.11 × 10^−5^ *	2.12 *	2.18 **	0.21	−2.25
TIMP-3	93	5.68 × 10^−5^	1.87 *	2.02 **	0.25 *	−2.22
TIMP-4	93	4.30 × 10^−4^	1.57	1.86 **	0.23	−2.08
TNF-α	91	−2.89 × 10^−2^	2.57 *	1.91 **	0.27 *	−2.47
TNF-β	65	−1.53 × 10^−2^	3.36 *	3.03 **	0.11	−4.48 *
TREM-1	85	9.48 × 10^−5^	1.18	1.60 *	0.22	−1.39
VEGF	93	6.03 × 10^−4^	1.49	1.79 *	0.23	−1.94
GM-CSF ^f^	41	−	−	−	−	−
IL-17 ^f^	13	−	−	−	−	−
TGF-α ^f^	32	−	−	−	−	−

ADAMTS13, a disintegrin and metalloproteinase with a thrombospondin type 1 motif member 13; BMP, bone morphogenetic protein; CIA, chronic inflammatory arthritis; Coef., coefficient; EGF, epidermal growth factor; FAP, fibroblast activation protein; FGF, fibroblast growth factor; G-CSF, granulocyte-colony stimulating factor; GM, granulocyte-macrophage; HGFR, hepatocyte growth factor receptor; ICAM, intercellular adhesion molecule; IL, interleukin; IP, interferon gamma-induced protein; MCP, monocyte chemoattractant protein; MIP, macrophage inflammatory protein; MMP, matrix metalloproteinase; NCAM, neural cell adhesion molecule; No., number; obs., observations; OPG, osteoprotegerin; palp., palpation; PDGF, platelet-derived growth factor; TIMP, tissue inhibitors of metalloproteinases; TGF, transforming growth factor; TNF, tumour necrosis factor; TREM, triggering receptor expressed on myeloid cells; VEGF, vascular endothelial growth factor. ^a^ The ordered logistic regression was modelled from successful outcome in three steps down to intermediate/deteriorated outcome as the worst outcome. A positive coefficient thereby indicating that the higher the specific protein concentration, the worse the outcome. ^b^ A positive coefficient shows that the diagnosis or variable affects the outcome in a negative way. ^c^ Describes the interaction between CIA and positive jaw muscle palpation related to outcome. A negative coefficient indicates a positive correlation to outcome. ^d^ No observations in the sample. ^e^ Omitted because of collinearity. ^f^ Too few observations why calculations could not be done. * *p* ≤ 0.05, ** *p* ≤ 0.005.

## Data Availability

The data presented in this study are available on request from the corresponding author.

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
