# Peer review of "Synovial Tissue Proteins and Patient-Specific Variables as Predictive Factors for Temporomandibular Joint Surgery"

_diagnostics, 2020, doi:10.3390/diagnostics11010046_

Round 1

Reviewer 1 Report

The paper Synovial tissue proteins and patient-specific variables as predictive factors for temporomandibular joint surgery by Ulmner et al., is interesting and well written. This paper has some limitations which the authors point out. Namely, a short follow-up period and modest correlation of proteins with outcome. The TMJ cartilage is somewhat unique compared to some other joints, and has been shown to have some ability to self-repair. It would have been nice to have the authors comment on that in the discussion and how it might be reflected in the modest correlations. 

It is unfortunate that the sex ratio was not more balanced in all groups but especially OA and CIA.

For me, the bar graphs in Figure 3 were difficult to process. I wonder if the authors considered expressing the data in linear form.

Although the correlation value was modest for Syndecan-1 it was a little puzzling that the authors didn't include it in their discussion. It has previously been shown to be associated with attempted cartilage repair.

Reviewer 2 Report

Dear Authors,

I've really appreciated Your manuscript and the topic of clinical interest.

The article has been well conducted and it does not need any changes exept for some English grammar corrections and MDPI formatting style adaptations.

Good Job.

Thank You

Kind Regards
